# Trust Formation, Error Impact, and Repair in Human–AI Financial Advisory: A Dynamic Behavioral Analysis

**DOI:** 10.3390/bs15101370

**Published:** 2025-10-07

**Authors:** Jihyung Han, Daekyun Ko

**Affiliations:** 1Research Institute of Human Ecology, Seoul National University, Seoul 08826, Republic of Korea; ellyhan@snu.ac.kr; 2Department of Consumer Science and Glocal Life-Care Convergence, Chungnam National University, Daejeon 34134, Republic of Korea

**Keywords:** artificial intelligence, trust dynamics, temporal processes, algorithmic trust, error sensitivity, explainable AI

## Abstract

Understanding how trust in artificial intelligence evolves is crucial for predicting human behavior in AI-enabled environments. While existing research focuses on initial acceptance factors, the temporal dynamics of AI trust remain poorly understood. This study develops a temporal trust dynamics framework proposing three phases: formation through accuracy cues, single-error shock, and post-error repair through explanations. Two experiments in financial advisory contexts tested this framework. Study 1 (N = 189) compared human versus algorithmic advisors, while Study 2 (N = 294) traced trust trajectories across three rounds, manipulating accuracy and post-error explanations. Results demonstrate three temporal patterns. First, participants initially favored algorithmic advisors, supporting “algorithmic appreciation.” Second, single advisory errors resulted in substantial trust decline (η^2^ = 0.141), demonstrating acute sensitivity to performance failures. Third, post-error explanations significantly facilitated trust recovery, with evidence of enhancement beyond baseline. Financial literacy moderated these patterns, with higher-expertise users showing sharper decline after errors and stronger recovery following explanations. These findings reveal that AI trust follows predictable temporal patterns distinct from interpersonal trust, exhibiting heightened error sensitivity yet remaining amenable to repair through well-designed explanatory interventions. They offer theoretical integration of appreciation and aversion phenomena and practical guidance for designing inclusive AI systems.

## 1. Introduction

As artificial intelligence becomes increasingly embedded in decision-making contexts—from financial advisory to healthcare diagnosis—a fundamental behavioral question emerges: how do people develop, maintain, and adjust their trust in AI systems over time? This question extends beyond the well-documented phenomenon of initial technology acceptance to encompass the more complex dynamics of trust evolution through repeated human–AI interactions ([17]). Understanding these temporal dynamics is critical because trust serves as a primary gateway through which AI technologies either succeed in augmenting human capabilities or fail to realize their potential benefits.

The behavioral stakes of this inquiry are substantial. Unlike human-to-human trust, which develops through rich social and emotional exchanges, AI trust appears to follow distinct psychological pathways characterized by systematic evaluation of performance signals and cognitive processing of system explanations ([33]; [39]). This cognitive-centric nature of AI trust makes it potentially more volatile than interpersonal trust, as users may form strong initial expectations about AI reliability that become quickly disrupted by observable errors ([9]). However, the same cognitive foundation may also make AI trust more amenable to deliberate repair through well-designed explanatory interventions—a possibility with profound implications for human–AI collaboration.

Recent research has begun to map the landscape of human-AI trust, revealing contrasting patterns of “algorithm aversion” (reluctance to rely on AI after observing errors) and “algorithm appreciation” (preference for AI consistency over human judgment; [28]). However, these findings primarily stem from single-interaction studies that cannot capture how trust unfolds across repeated encounters. More critically, existing work has not sufficiently theorized the temporal mechanisms through which AI trust develops and potentially recovers after disruption, leaving unresolved the apparent contradiction between appreciation and aversion phenomena.

This theoretical gap is particularly pronounced in consequential decision-making contexts where trust dynamics have significant practical implications. While laboratory studies have established that both algorithm appreciation and aversion can occur, the conditions under which each pattern emerges—and how they might coexist within temporal sequences—remain poorly understood. This represents a significant limitation in our understanding of human behavior in AI-enabled environments, particularly given that real-world AI interaction involves repeated encounters where initial impressions can be confirmed, violated, or repaired through ongoing performance feedback and system responses.

The financial advisory domain provides an ideal context for examining these dynamics of trust. Financial decision-making represents a high-stakes context where trust is essential for effective human–AI collaboration, yet it is also a context where performance feedback is readily observable and interpretable. Moreover, financial advisory exemplifies a broader class of AI applications where users must decide whether to accept algorithmic recommendations that have significant personal consequences. The domain exhibits optimal conditions for studying algorithm appreciation, characterized by quantitative task demands, substantial expertise gaps between users and the required competence, and delayed performance feedback that prevents immediate error detection.

Understanding trust dynamics in this domain thus has implications extending beyond financial services to encompass any context where AI systems provide consequential advice or recommendations. The financial context also enables a systematic investigation of individual differences, particularly financial literacy, which emerges as a focal moderator that may systematically condition temporal trust dynamics through an enhanced capacity to process performance signals and explanatory information. Financial literacy equips users with probabilistic reasoning skills and risk assessment capabilities, which are essential for calibrating AI trust appropriately across varying performance conditions—a critical factor in designing inclusive AI systems that effectively serve diverse user populations.

Building on trust development theory ([33]) and the emerging literature on explainable AI, we propose a temporal trust dynamics framework that explains how AI trust evolves through three predictable behavioral stages. The first stage involves forming trust through accuracy-based competence assessments, where users develop initial trust based on their perception of the system’s performance and reliability. The second stage encompasses a single-error shock, characterized by a disproportionate decline in trust following observable system failures that violate established performance expectations. The third stage involves post-error repair through explanatory transparency, where well-designed explanations can clarify system limitations and provide actionable guidance for future interactions. This repair process may extend beyond simple recovery to potentially strengthen trust through structured exposure to systematic advisory processes, particularly in short-term, structured interactions such as simulated financial advisory tasks, creating opportunities for enhanced human-AI collaboration even following performance failures.

This framework makes several novel contributions to the field of behavioral science. First, it provides the first systematic temporal account of AI trust development, moving beyond static adoption models to dynamic process theories that can predict how trust evolves through repeated interactions. Second, it identifies specific behavioral mechanisms—particularly the role of explanatory transparency in trust repair—that have both theoretical significance and practical implications for the design of AI systems. Third, it bridges multiple theoretical streams, including trust development theory, algorithm evaluation research, and explainable AI, into a unified behavioral framework that can guide both research and practice.

We test this framework through two complementary experiments in financial advisory contexts. Study 1 establishes baseline trust patterns through systematic comparison of human versus algorithmic advisors under varying risk conditions, providing foundational evidence for initial trust formation mechanisms and testing whether the financial context indeed favors algorithm appreciation. Study 2 traces trust trajectories across three interaction rounds, manipulating both advisory accuracy and post-error explanatory transparency to test the complete temporal dynamics proposed by our framework.

Together, these studies provide the first systematic behavioral account of how AI trust develops, deteriorates, and potentially recovers through repeated interactions. The findings reveal that successful human–AI collaboration depends not on eliminating errors, but on designing systems that can navigate the predictable dynamics of trust formation, disruption, and repair that characterize human responses to algorithmic advice. These insights have broader implications for AI system design and governance research that can foster appropriate trust calibration across diverse user populations.

## 2. Literature Review and Hypotheses

### 2.1. Cognitive and Behavioral Foundations of AI Trust

Trust in artificial intelligence represents a unique form of human cognition that bridges traditional interpersonal trust mechanisms with technology acceptance behaviors. Unlike interpersonal trust, which develops through complex social exchanges involving emotional reciprocity and shared experiences ([38]), AI trust appears fundamentally grounded in cognitive assessments of system competence, reliability, and predictability ([27]; [33]). Research in human–automation interaction has established that trust in automation functions as a quantifiable attitude that determines the extent of reliance on automated agents, constituting a psychological state that directly influences behaviors ([16]; [27]).

This cognitive-centric nature creates both opportunities and vulnerabilities for the development of trust. Humans tend to interact with people and machines in somewhat similar, yet not identical, ways, with automated decision aids often designed to emulate human interactions, despite users remaining conscious that computers lack human attributes ([30]). This creates a unique psychological context where users must navigate between anthropomorphic responses and rational performance evaluation.

From a behavioral decision-making perspective, cognitive load theory provides crucial insights into AI trust mechanisms. Research demonstrates an inversely proportional relationship between trust and cognitive load, where increased cognitive demands reduce trust ratings in human-automation interactions ([1]). When cognitive resources are constrained, users may default to simplistic reliance strategies, such as indiscriminately following AI recommendations ([8]). This cognitive efficiency principle helps explain why AI trust exhibits greater volatility compared to interpersonal trust, as users rely on simplified heuristics when evaluating AI performance under resource limitations.

The cognitive foundation of AI trust makes it susceptible to systematic biases. Confirmation bias is particularly evident in AI-assisted decision-making, where users tend to show increased trust and acceptance when AI recommendations align with their initial judgments ([7]). Additionally, mere knowledge that advice originates from AI lead to cause overreliance, causing users to follow AI recommendations even when they contradict available contextual information ([26]). This overreliance phenomenon represents a systematic departure from optimal decision-making, highlighting the need for interventions that promote appropriate trust calibration.

Perhaps most critically, AI trust exhibits asymmetric temporal dynamics. Building trust in AI takes more time compared to building trust in humans; however, when AI encounters problems, trust loss occurs more rapidly, with simpler tasks showing greater degradation of trust following errors ([12]; [31]). This pattern—slow trust building, rapid trust loss—reflects the cognitive-evaluative nature of AI trust, where negative performance signals carry disproportionate weight in trust updating processes.

These cognitive-evaluative characteristics create context-dependent variations in the initial formation of trust. While algorithm appreciation has been demonstrated across various domains ([28]), systematic evidence remains limited for high-stakes analytical contexts with delayed performance feedback. The financial advisory domain converges on boundary conditions that amplify algorithm appreciation: quantitative task demands, substantial expertise gaps between users and the required competence, and the absence of immediate performance feedback, which typically triggers aversion responses ([6]; [10]). This confluence of analytical task demands, expertise gaps, and delayed feedback represents an ideal test case for examining how contextual factors shape initial algorithmic preference and provides a foundation for understanding trust evolution through repeated interactions.

Given these contextual advantages for algorithmic competence perception in financial decision-making—quantitative task demands, limited user expertise for direct evaluation, and the absence of immediate performance feedback—we predict an initial preference for algorithms over human advisory services across trust, satisfaction, and reliance measures.

**H1.** 
*Relative to human advisors, algorithmic advisors will yield higher (a) trust, (b) satisfaction, and (c) reliance intention.*


### 2.2. Algorithm Appreciation vs. Algorithm Aversion: Reconciling Contradictory Behavioral Patterns

The literature on human responses to algorithmic advice reveals seemingly contradictory behavioral patterns that require systematic reconciliation. Early research emphasized algorithm aversion—the tendency to reject algorithmic recommendations more readily than equivalent human advice, particularly after observing errors ([9], [10]). However, subsequent investigations have documented algorithm appreciation—consistent preferences for algorithmic over human judgment across various analytical tasks ([28]). Understanding when and why each pattern emerges is crucial for predicting human behavior in AI-enabled decision contexts.

Algorithm aversion represents a systematic departure from rational decision-making where users reject superior algorithmic advice in favor of inferior human alternatives. This phenomenon appears rooted in several cognitive and motivational mechanisms. First, users often maintain unrealistic expectations for algorithmic perfection—what researchers term “perfect automation schema”—leading them to interpret any error as evidence of fundamental system inadequacy rather than normal performance variation ([9]). This contrasts sharply with tolerance for human errors, which are more readily attributed to situational factors.

Research demonstrates that algorithm aversion exhibits specific behavioral signatures. Even when users observe algorithmic predictions outperforming human predictors, they still tend to resist algorithmic advice, indicating that aversion persists beyond mere performance concerns ([10]). A systematic review of 80 empirical studies reveals that algorithm aversion manifests across diverse domains and is influenced by algorithm characteristics, individual differences, task factors, and higher-level contextual elements ([32]). Notably, the relationship between trust and automation reliance shows that users often exhibit disuse patterns where they perform tasks manually to a greater extent than warranted, resulting in suboptimal joint human–machine performance ([3]).

Contrary to aversion findings, research on algorithm appreciation demonstrates consistent preferences for algorithmic advice across various contexts. [28] ([28]) found that people give more weight to identical advice when labeled as algorithmic versus human in origin, a pattern observed across visual estimates, geopolitical forecasting, business predictions, and social judgments. This appreciation appears particularly pronounced in analytical tasks where objectivity and consistency are valued over subjective judgment.

Recent work reveals that these contradictory patterns can be reconciled through understanding how agent framing affects perceived competence and temporal dynamics. A systematic review by [5] ([5]) highlighted that algorithm aversion and appreciation can both emerge within similar paradigms depending on how human versus algorithmic agents are framed, with effects mediated by perceived competence or expert power. This suggests that both phenomena reflect rational responses to perceived agent capabilities rather than fixed psychological biases.

The temporal aspect of algorithm preference proves critical for reconciling these contradictory patterns. Algorithm appreciation may dominate initial encounters when users lack performance feedback and must rely on general competence expectations, while algorithm aversion emerges following error observations that violate established expectations ([6]). This temporal perspective suggests that both phenomena represent adaptive responses to available information, mediated by cognitive biases and contextual factors rather than competing psychological tendencies.

Several factors moderate the direction and magnitude of algorithm preference patterns. Task characteristics play a crucial role, with algorithm appreciation being stronger in quantitative and analytical tasks and weaker in subjective and creative domains ([6]). Individual differences in uncertainty tolerance and analytical thinking—not merely task characteristics—better predict patterns of algorithmic reliance ([40]). Cognitive load emerges as another critical moderator, with increased cognitive demands strengthening reliance on algorithmic advice, particularly under conditions of task complexity and time pressure ([47]).

### 2.3. Single-Error Shock and Error Tolerance in Algorithmic Contexts

A central controversy in the human-AI interaction literature concerns users’ reactions to algorithmic errors, particularly the phenomenon of disproportionate trust decline following single failures. The algorithm aversion perspective suggests that even one visible mistake can lead to significant decreases in perceived competence and willingness to rely on algorithmic systems ([9], [10]). This pattern reflects what researchers term “perfect automation schema”—users’ tendency to expect flawless performance from AI systems and interpret any deviation as evidence of fundamental inadequacy rather than normal performance variation.

Empirical evidence for single-error sensitivity is robust across domains. [9] ([9]) found that participants avoided algorithmic forecasters more readily than human forecasters after observing identical prediction errors, even when the algorithm demonstrably outperformed human alternatives. This aversion persisted despite explicit information about superior algorithmic accuracy, suggesting that error observations trigger cognitive responses that override rational performance assessment. Subsequent work has documented similar patterns in medical diagnosis ([18]), investment advice ([25]), and consumer product recommendations ([6]), indicating that single-error shock represents a general psychological phenomenon rather than domain-specific bias.

However, the temporal dynamics of error tolerance reveal important nuances that challenge simplistic aversion accounts. Research in human-automation interaction reveals that error sensitivity varies systematically in relation to performance expectations, task characteristics, and feedback timing ([27]; [31]). When users develop strong competence priors through initial positive experiences, subsequent errors may trigger a sharper decline in trust than would occur in the absence of such priors—a pattern consistent with expectancy violation theory from social psychology ([20]). This suggests that a single-error shock may be most pronounced precisely in contexts where initial algorithm appreciation has established high performance expectations.

The perfect automation schema also exhibits systematic boundary conditions across individual and cultural factors. Research demonstrates variation in schema strength based on cultural technology trust, analytical thinking capabilities, and domain expertise ([17]; [40]). These moderating factors suggest that single-error sensitivity represents a general yet contextually variable phenomenon that requires systematic investigation.

The automation bias literature adds complexity by documenting oscillations between over-reliance and under-reliance as users process performance feedback over time. Users often exhibit miscalibration relative to systems’ actual capabilities, alternating between uncritical acceptance and excessive skepticism depending on the salience of recent experience ([18]). These oscillations suggest that single-error effects may represent acute but potentially recoverable disruptions to trust, rather than permanent rejection, creating opportunities for systematic repair interventions.

Building on this evidence, we hypothesize that single-error shocks operate through a competence-based trust updating mechanism. When an AI system that has established positive competence expectations commits a visible error, users should engage in rapid revision that sharply downgrades trust, satisfaction, and behavioral reliance. This response should be more pronounced for AI than for human advisors because users may attribute human errors to situational factors, while interpreting AI errors as systematic limitations that reflect fundamental model inadequacy.

**H2.** 
*A single inaccuracy will reduce (a) trust, (b) satisfaction, and (c) reliance intention relative to accurate advice.*


### 2.4. Explanatory Transparency as a Post-Error Trust Repair Mechanism

If errors are inevitable in AI systems, the critical behavioral question becomes whether—and how—explanatory transparency can facilitate trust repair following performance failures. The explainable AI (XAI) literature reveals that transparency is not uniformly beneficial; explanations can impose cognitive load, distract attention from relevant cues, or even inflate misplaced confidence, depending on their design, timing, and user characteristics ([35]; [37]; [41]). This complexity necessitates a nuanced understanding of when and how explanations support appropriate trust calibration rather than blind acceptance or wholesale rejection.

The effectiveness of explanations is mechanism-specific, as shown in the trust repair literature from interpersonal contexts, which provides a theoretical foundation for understanding post-error explanations. [24] ([24], [23]) demonstrate that effective trust repair requires interventions tailored to the type of violation: competence-based violations benefit from explanations that clarify causes and demonstrate corrective understanding, while integrity-based violations require different approaches. Translating this framework to AI contexts, algorithmic errors typically represent competence rather than integrity violations, suggesting that explanations should focus on clarifying system limitations and boundary conditions rather than offering generic acknowledgments.

Recent research identifies specific design principles for adequate post-error explanations in AI systems. Explanations that combine causal attribution (explaining why the error occurred) with boundary specification (identifying system limitations) prove most effective for competence-based trust repair ([36]). This “cause-limits” framework enables users to form more accurate mental models of system capabilities while managing cognitive load constraints inherent in post-error contexts.

The timing of explanatory interventions proves crucial for the effectiveness of repair. Pro-active explanations may establish general expectations but fail to address the specific attributional processes triggered by concrete failures ([22]). Post-error explanations, delivered immediately following performance failures, can target the precise causal inferences driving trust decline and provide contextually relevant information for trust recalibration. This timing advantage may be significant given evidence that trust updating occurs rapidly following error observations, creating narrow windows for interventional effectiveness.

Empirical evidence on the effectiveness of explanations in trust repair contexts reveals important boundary conditions. [43] ([43]) found that explanations enhanced trust and acceptance in AI systems, but primarily when they were perceived as causally informative rather than merely descriptive. Similarly, explanations improve user decision quality when they provide actionable insights rather than technical details that impose additional cognitive burden without supporting behavioral adjustment ([4]).

Individual differences in explanation processing capacity suggest heterogeneous repair effects. Users with higher domain expertise may derive greater value from detailed explanations because they possess the necessary background knowledge to interpret causal information and translate it into appropriate behavioral adjustments ([14]). Conversely, users with limited domain knowledge may benefit more from explanations that focus on behavioral implications rather than technical mechanisms.

Building on this theoretical foundation, we predict that post-error explanations will function as a behavioral repair mechanism by clarifying system limitations and providing contextual guidance for future interactions. The present study examines the basic effectiveness of post-error explanations (provided vs. not provided) rather than comparing specific explanation types. While research suggests that explanations combining causal attribution with boundary specification may be most effective (cause-limits framework), systematic comparison of explanation types represents an important direction for future research. Our focus on the presence versus absence of explanation provides foundational evidence for whether transparency interventions can facilitate trust repair following algorithmic errors.

**H3.** 
*Post-error explanations will attenuate the negative effects of inaccuracy on (a) trust and (b) reliance intention relative to no explanation.*


### 2.5. Individual Differences and the Role of Context: Financial Literacy as Focal Moderator

Users differ systematically in how they interpret performance signals and integrate explanatory content during human–AI advisory interactions, creating heterogeneous patterns in trust formation, error sensitivity, and repair effectiveness. While the technology acceptance literature has established broad relationships between individual characteristics and AI adoption intentions ([15]; [45]), the temporal dynamics of trust evolution remain underexplored from a perspective of individual differences. Understanding these heterogeneous patterns is crucial because the same AI system may produce markedly different user experiences depending on individual cognitive capabilities and contextual knowledge.

Prior work has linked general technology readiness and AI literacy to stronger intentions to use AI-mediated services across various domains ([46]). However, these broad predispositions may be less informative for understanding temporal trust dynamics than domain-specific competencies that directly influence users’ ability to evaluate performance feedback and process explanatory information. In financial advisory contexts, financial literacy emerges as a particularly relevant moderator because it equips users with the conceptual tools necessary to assess probabilistic recommendations, recognize boundary conditions, and distinguish systematic patterns from random variation.

Financial literacy influences trust dynamics through three mechanisms: enhanced signal detection, error contextualization, and explanation processing capabilities ([14]; [29]). More literate users can distinguish meaningful performance patterns from noise, contextualize single errors within broader performance trends, and translate explanatory content into behavioral adjustments. This enhanced processing creates differential sensitivity to performance feedback throughout trust formation, error response, and repair phases.

While primarily of theoretical interest, literacy-based moderation also suggests secondary implications for consumer protection in AI-mediated financial services. Understanding how individual differences affect AI trust patterns may inform future policy considerations, though such applications remain beyond the scope of the present theoretical investigation.

Building on these considerations, we predict that financial literacy will moderate temporal trust trajectories across repeated advisory interactions. This study focuses on literacy as the focal individual difference while acknowledging that cultural factors, cognitive load effects, and prior AI experience represent important boundary conditions for future systematic investigation.

**H4.** 
*Financial literacy moderates the temporal trajectory of trust and reliance during repeated advisory interactions, such that higher literacy increases sensitivity to performance feedback across rounds (i.e., sharper decline after a single error and greater overall recovery across subsequent rounds).*


We distinguish this focal prediction from related constructs examined as covariates. Baseline trust in AI represents a stable individual predisposition that should influence overall trust levels across interactions but not necessarily moderate temporal dynamics ([17]). Product risk preferences have been proposed as moderators of advisor type effects, but empirical evidence remains inconsistent and appears sensitive to task structure and information design ([6]; [21]). Given mixed findings, we examine risk effects exploratorily without formal predictions.

### 2.6. Conceptual Framework

Figure 1 synthesizes the proposed temporal trust dynamics framework, integrating the three theoretical phases with their corresponding behavioral mechanisms and moderating factors. The framework specifies trust evolution as a sequential process where initial algorithm appreciation (Stage 1) establishes performance expectations that amplify subsequent error sensitivity (Stage 2). At the same time, post-error explanations create opportunities for trust recalibration and partial recovery (Stage 3).

The model positions financial literacy as the focal individual difference moderator that conditions trust trajectories across all three stages, reflecting enhanced capacity to process performance feedback and explanatory information. Baseline trust in AI operates as a stable predisposition that influences overall trust levels, rather than temporal dynamics, while product risk is examined exploratorily due to inconsistent prior findings. This specification maintains focus on the accuracy → error → explanation sequence while acknowledging individual and contextual factors that shape response magnitude and duration.

The framework advances existing trust theories by integrating temporal dynamics with cognitive-evaluative mechanisms specific to AI systems. Unlike interpersonal trust models that emphasize social and emotional exchange processes, this framework centers on performance-based competence assessment as the primary driver of trust evolution. The sequential nature of the model—where each stage influences subsequent phases—reflects the path-dependent character of human–AI interaction, where early experiences establish expectations that fundamentally shape responses to later events.

This integrated framework directly informs our four hypotheses: H1 examines initial algorithm appreciation (Stage 1), H2 tests single-error shock mechanisms (Stage 2), H3 evaluates post-error repair through explanations (Stage 3), and H4 investigates how financial literacy moderates these temporal patterns across all stages. Together, these hypotheses provide a comprehensive test of the proposed temporal trust dynamics model depicted in Figure 1.

## 3. Materials and Methods

We conducted two complementary experiments that align with the temporal framework shown in Figure 1. Study 1 employed a 2 (advisory type: human vs. algorithmic) × 2 (product risk: low vs. high) between-subjects design to test H1 at Round 1 and explore risk moderation. Study 2 adopted a mixed design across three rounds to test H2 (single-error shock at Round 2) and H3 (post-error repair via explanations from Round 2 → 3), and to examine H4 (financial literacy as a moderator of the Round 1 → 3 trajectory). Primary outcomes were trust, satisfaction, and reliance intention (interpretive label for the scale commonly termed continued usage intention).

### 3.1. Common Materials, Measures, and Procedures

**Participants and recruitment.** Both studies recruited participants from Embrain’s panel database, comprising over 1.5 million members in South Korea. Eligible respondents were adults aged 20–50 with experience in fund investments. Pilot testing preceded each study to confirm the clarity and reliability of the survey (Study 1: N = 40, 1–2 July 2025; Study 2: N = 60, 9–10 July 2025), resulting in minor procedural adjustments. Primary data collection took place from 4 July to 9 July 2025 (Study 1) and 14–17 July 2025 (Study 2). Participants provided informed consent and received KRW 4000 (~USD 3–4) compensation following standard practices for online survey participation.

**Experimental stimuli and administration.** All scenarios and experimental manipulations (advisory type, product-risk framing, single-error message, post-error explanation) were developed and implemented by the authors (details in Appendix A). Advisory-type manipulations presented recommendations as either “based on analysis by certified financial experts with 10+ years experience” (human condition) or “generated by an AI-based robo-advisor system using advanced algorithms” (algorithmic condition), while holding recommendation content identical. Product risk was varied through financial product descriptions pre-tested for risk perception differences. Post-error explanations provided a detailed analysis of investment losses, including sector-specific factors (e.g., the decline of the biotech sector, global interest rate impacts, and concentration risk), emphasizing the importance of diversification and risk management for future investment decisions. The survey was administered using Embrain’s online platform with automated random assignment to experimental conditions.

**Primary outcome measures.** Trust, satisfaction, and reliance intention were assessed using established scales adapted from prior research. Trust employed a 5-item scale measuring perceived reliability, competence, and benevolence ([2]). Satisfaction used a single-item global evaluation. Reliance intention was measured using a 3-item scale assessing continued usage intentions. All measures employed 7-point Likert scales, ranging from 1 (strongly disagree) to 7 (strongly agree). Scale reliabilities demonstrated good internal consistency across both studies and measurement rounds (details in Appendix A).

**Individual differences and control measures.** Financial literacy was measured using a 6-item short form assessing numeracy and understanding of financial concepts. This scale was chosen for its brevity and validation in Korean populations. It was treated as a continuous variable based on summed correct responses (range: 0–6) to preserve statistical power for moderation analyses. Additional measures included financial skill (9-item CFPB scale), self-efficacy (5-item scale adapted from [42]), AI literacy (5-item scale from [44]), and baseline trust in AI (3-item scale from [19]). Risk attitude was assessed using a single-item measure from [11] ([11]) on an 11-point scale. Investment-related behaviors included investment independence and patterns of prospectus review. All scales demonstrated adequate reliability (Appendix A).

**Manipulation checks.** Systematic checks assessed: (i) perceived advisor identity (human vs. algorithmic); (ii) perceived product risk (Study 1); (iii) recognition of Round 2 inaccuracy (Study 2); and (iv) receipt of post-error explanations (Study 2). All manipulations were successfully implemented (detailed results are provided in Appendix A).

**Data processing and analysis approach.** Randomization was executed automatically by the Embrain platform. Data cleaning was performed according to pre-specified protocols, excluding only responses with logical inconsistencies or extreme outliers. No missing data occurred on primary variables; all analyses were conducted on complete cases. Sample size determination used G*Power (version 3.1; [13]) calculations targeting small-to-medium effects (f = 0.15, α = 0.05, power = 0.80), requiring minimum N = 180 for Study 1 and N = 280 for Study 2, with recruitment continuing until minimum thresholds were achieved. The final samples (N = 189, N = 294) exceeded the required thresholds. Statistical analyses were IBM SPSS Statistics 30.0 using ANOVA/ANCOVA for Study 1 and repeated-measures analyses for Study 2. Linear mixed-effects models with participant-level random intercepts were planned as robustness checks to account for individual differences in baseline trust levels and validate key interaction effects.

### 3.2. Study 1: Participants, Design, Materials, and Procedure

**Participants and design.** Study 1 employed a 2 (advisory type: human expert vs. AI robo-advisor) × 2 (product risk: low vs. high) between-subjects factorial design to test H1 and explore risk moderation effects. Following pilot testing (N = 40) to confirm survey clarity, 200 participants were recruited from the Embrain panel using stratified sampling. After screening for invalid responses, the final sample comprised N = 189 participants (M_age = 39.96 years, 50.3% male). Random assignment across the four experimental conditions confirmed no significant demographic differences across groups (Table 1, Panel A).

**Materials and manipulations.** The advisory type was manipulated through advisor descriptions, while holding the recommendation content constant. Human expert conditions presented advice attributed to certified financial advisors with relevant credentials and experience. AI robo-advisors provide identical recommendations as algorithm-generated advice from an automated financial advisory system. Product risk was varied through financial product descriptions that were pre-tested for differences in risk perception. Low-risk scenarios featured conservative fund products that emphasized capital preservation and stable returns, while high-risk scenarios presented volatile products with substantial return potential, alongside significant loss risks.

**Procedure.** Participants completed a standardized sequence: (1) informed consent and demographic collection, (2) random assignment to experimental conditions, (3) financial scenario review and product evaluation based on assigned risk level, (4) advisory recommendation reception according to assigned advisor type, (5) completion of manipulation checks and control variable measures, (6) assessment of primary outcomes (trust, satisfaction, reliance intention), and (7) compensation and debriefing. Sessions lasted approximately 10 to 15 min.

**Analysis plan.** H1 was tested using two-way ANOVAs on each outcome variable with advisory type and product risk as fixed factors. Risk moderation effects were examined exploratorily without formal predictions. Robustness analyses employed ANCOVAs including baseline trust in AI and financial literacy as covariates to control for individual predispositions. The effectiveness of manipulation checks was verified using chi-square tests for categorical perceptions and independent samples t-tests for continuous measures.

### 3.3. Study 2: Participants, Design, Materials, and Procedure

**Participants and design.** Study 2 employed a mixed design with Round (1, 2, or 3) as a within-subjects factor and two between-subjects factors: Accuracy (accurate vs. single inaccuracy at Round 2) and Post-error Explanation (provided vs. not provided between Rounds 2 and 3). Following pilot testing (N = 60) to verify procedural clarity, 300 participants were recruited using the same eligibility criteria as those in Study 1. After excluding incomplete responses, the final sample comprised N = 294 participants (M_age = 39.86 years, 49.7% male) randomly assigned across six experimental conditions with balanced demographic characteristics (Table 1, Panel B).

**Materials and manipulations.** The study employed a three-round advisory interaction sequence. Round 1 delivered accurate recommendations across all conditions to establish initial performance baselines. In Round 2, the accuracy manipulation was implemented: control conditions received continued accurate advice, while experimental conditions received intentionally inaccurate recommendations followed by explicit feedback revealing the prediction error. Performance feedback was provided through visual displays of actual investment returns (showing negative performance), accompanied by messages such as “Your selected ‘Blue Wave Fund’ has shown results different from expectations (negative). A comprehensive strategy review is needed to achieve your goal of $15 million for medical school tuition.” Between Rounds 2 and 3, participants in the explanation conditions received brief post-error explanations that specified the cause of the inaccuracy, system limitations, and guidance for interpreting subsequent recommendations. These explanations followed the “cause-limits“ framework identified in the literature review.

**Procedures.** Participants completed three consecutive rounds of financial advisory scenarios according to their assigned conditions. Each round followed an identical structure: scenario presentation, advisor recommendation reception, outcome feedback (accurate/inaccurate), and completion of trust, satisfaction, and reliance measures. In Round 2, participants assigned to the inaccuracy conditions received deliberately incorrect recommendations accompanied by clear performance feedback. Those in the explanation conditions then received explanatory transparency before proceeding to Round 3. Manipulation checks were administered after each round to verify successful implementation. Sessions lasted approximately 15 to 20 min.

**Analysis plan.** H2 and H3 were tested using repeated-measures ANOVAs with Round as the within-subjects factor and Accuracy/Explanation as the between-subjects factor. The focal tests examined Round × Accuracy interactions (single-error shock, H2) and Round × Explanation interactions (post-error repair, H3). H4 was tested by adding financial literacy as a continuous moderator in mixed-effects models examining Round × Literacy interactions across the complete trajectory. Linear mixed-effects models with participant-level random intercepts were employed as robustness checks, with more complex random slope specifications attempted but limited by convergence constraints. Baseline trust in AI was included as a covariate in robustness analyses. Greenhouse-Geisser corrections were applied when the sphericity assumption was violated.

### 3.4. Ethics, Data/Materials, and GenAI Use

**Ethical approval and consent.** The study was conducted in accordance with the Declaration of Helsinki and approved by the Seoul National University Institutional Review Board (IRB No. 2505/003-001; approved 30 June 2025). Written informed consent was obtained from all participants prior to their participation in the study. Participants received monetary compensation (KRW 4000) upon completion, in line with standard practices for online survey research in South Korea.

**Data/Materials.** Data are not publicly hosted due to privacy considerations; all de-identified data and experimental materials will be provided upon reasonable request to the corresponding author. Complete experimental stimuli and measurement instruments are available in the Appendix A.

**Generative AI disclosure.** Generative AI tools were used solely for language polishing and editing assistance. No generative AI was used in the study design, data generation, statistical analysis, or interpretation of the results. All substantive content, theoretical development, and empirical conclusions were developed and verified by the authors.

## 4. Results

### 4.1. Study 1

**Descriptive statistics and manipulation checks.** The final sample comprised N = 189 participants, approximately balanced across the four experimental conditions (43/50/48/48). Demographic characteristics showed no significant differences across conditions, confirming successful randomization (Table 1, Panel A). Manipulation checks verified effective implementation of both experimental factors: participants accurately recognized product risk assignments (χ^2^ = 127.506, *p* < 0.001) and advisory type conditions (χ^2^ = 104.359, *p* < 0.001), with detailed results reported in Appendix A.

**H1: Initial algorithmic advantage.** H1 predicted that algorithmic advisors would receive higher initial trust, satisfaction, and reliance ratings compared to human experts. Two-way ANOVAs revealed consistent support across all outcome measures. For trust, there was a significant main effect of advisory type, F (1,185) = 7.820, *p* = 0.006, η^2^ = 0.041, 95% CI [0.003, 0.095], with algorithmic advisors (M = 4.79, SD = 0.82) receiving higher ratings than human experts (M = 4.44, SD = 0.89). Similarly, algorithmic advisors received higher satisfaction ratings, F (1,185) = 5.507, *p* = 0.020, η^2^ = 0.029, 95% CI [0.001, 0.082] (AI: M = 4.89, SD = 0.96; Human: M = 4.53, SD = 1.15), and reliance intention, F (1,185) = 11.726, *p* < 0.001, η^2^ = 0.060, 95% CI [0.014, 0.126] (AI: M = 4.63, SD = 0.92; Human: M = 4.12, SD = 1.13). These results provide strong support for H1, demonstrating initial algorithm appreciation in financial advisory contexts (see Table 2 for detailed results).

**Exploratory analysis: Risk level effects.** Contrary to expectations from prior literature, product risk level showed no significant main effects across any outcome measures: trust F (1,185) = 0.859, *p* = 0.355, η^2^ = 0.005, 95% CI [0.000, 0.032]; satisfaction F (1,185) = 0.297, *p* = 0.586, η^2^ = 0.002, 95% CI [0.000, 0.018]; reliance F (1,185) = 0.607, *p* = 0.437, η^2^ = 0.003, 95% CI [0.000, 0.025]. Advisory type × risk interactions were also non-significant (all *p-values* > 0.190). Manipulation checks confirmed that participants correctly distinguished between low-risk and high-risk products (see Appendix A), indicating successful risk manipulation but minimal impact on advisor preference patterns. This suggests that the algorithmic advantage operates independently of product risk characteristics in initial encounters.

**Robustness analyses.** ANCOVAs including baseline trust in AI and financial literacy as covariates yielded identical qualitative patterns, confirming that advisory type effects were not attributable to individual predispositions. The algorithmic advantage remained significant for trust F (1,183) = 6.714, *p* = 0.010, η^2^ = 0.038, 95% CI [0.002, 0.088]; satisfaction F (1,183) = 5.338, *p* = 0.022, η^2^ = 0.030, 95% CI [0.001, 0.076]; and reliance F (1,183) = 10.485, *p* = 0.001, η^2^ = 0.057, 95% CI [0.010, 0.118].

### 4.2. Study 2

**Descriptive statistics and manipulation checks.** The final sample consisted of N = 294 participants, with balanced assignment across six experimental conditions (Table 1, Panel B). Manipulation checks verified successful implementation of all experimental factors. Participants accurately distinguished advisory types across all rounds (χ^2^ = 147.327, *p* < 0.001), correctly recognized the single inaccuracy at Round 2 (accurate conditions: M = 5.29; inaccurate conditions: M = 2.28; t = 20.556, *p* < 0.001), and appropriately identified explanatory transparency provision (χ^2^ = 27.070, *p* < 0.001). Full repeated-measures ANOVA results for Study 2 are summarized in Table 3.

**H2: Single-error shock.** H2 predicted that a single inaccuracy would reduce trust, satisfaction, and reliance intention relative to continued accurate advice. Repeated-measures ANOVAs revealed strong support through significant Round × Accuracy interactions. For trust, participants in the inaccuracy condition showed a marked decline from Round 1 to Round 2, compared to those receiving continued accurate advice, F (2,554) = 45.327, *p* < 0.001, η^2^ = 0.141, 95% CI [0.108, 0.171]. Similar patterns emerged for satisfaction, F (2,554) = 33.015, *p* < 0.001, η^2^ = 0.106, 95% CI [0.076, 0.135], and reliance intention, F (2,554) = 36.600, *p* < 0.001, η^2^ = 0.117, 95% CI [0.086, 0.146]. These large effect sizes demonstrate a disproportionate decline in trust following single errors, strongly supporting H2 and confirming the single-error shock phenomenon. The temporal patterns are illustrated in Figure 2, Panel A, and detailed in Table 3, Panel A.

**H3: Post-error repair through explanations.** H3 predicted that post-error explanations would attenuate the negative effects of inaccuracy on trust and reliance. Analysis of the explanation conditions (excluding accurate controls) revealed significant Round × Explanation interactions. Participants who received post-error explanations showed greater trust recovery from Round 2 to Round 3 compared to those who received no explanation, F (2,354) = 16.573, *p* < 0.001, η^2^ = 0.086, 95% CI [0.047, 0.129]. Similar repair effects emerged for satisfaction, F (2,354) = 6.709, *p* = 0.001, η^2^ = 0.037, 95% CI [0.009, 0.070], and reliance intention, F (2,354) = 4.079, *p* = 0.018, η^2^ = 0.023, 95% CI [0.002, 0.050]. These results support H3, demonstrating that explanatory transparency functions as an effective trust repair mechanism. The recovery patterns are shown in Figure 2, Panel B, and the corresponding ANOVA results appear in Table 3, Panel B.

**H4: Financial literacy moderation.** H4 predicted that financial literacy would moderate the temporal trajectory of trust and reliance, with higher literacy increasing sensitivity to performance feedback. This hypothesis received strong support through significant Round × Financial literacy interactions for trust, F (2,554) = 4.126, *p* = 0.017, η^2^ = 0.015, 95% CI [0.001, 0.038], and reliance intention, F (2,554) = 3.527, *p* = 0.030, η^2^ = 0.013, 95% CI [0.000, 0.035]. A similar trend emerged for satisfaction, F (2,554) = 2.100, *p* = 0.123, η^2^ = 0.008, 95% CI [0.000, 0.026]. Higher literacy participants exhibited sharper performance-based updating—both greater sensitivity to Round 2 errors and more pronounced recovery patterns across the complete trajectory, supporting H4 (see Table 3, Round × Financial literacy rows in Panels A and B).

**Round 3 recovery pattern analysis.** To determine whether trust recovery was complete or partial, we conducted paired t-tests comparing the levels of Round 1 and Round 3 levels within each condition. Results revealed an unexpected pattern of trust enhancement rather than mere recovery across all conditions. For participants who experienced errors followed by explanations, Round 3 trust levels (M = 4.75, SD = 0.85) were significantly higher than Round 1 baseline (M = 4.50, SD = 0.83), t (96) = −3.195, *p* = 0.002, d = −0.324, 95% CI [−0.528, −0.199]. Similarly, participants who experienced errors without explanations showed trust improvement from Round 1 (M = 4.55, SD = 1.00) to Round 3 (M = 4.73, SD = 1.04), t (96) = −1.936, *p* = 0.056, d = −0.197, 95% CI [−0.397, 0.005], though this was only marginally significant. Most remarkably, participants in the accurate condition demonstrated substantial trust enhancement from Round 1 (M = 4.60, SD = 0.81) to Round 3 (M = 4.88, SD = 1.12), t (99) = −3.547, *p* < 0.001, d = −0.355, 95% CI [−0.556, −0.152].

**Alternative explanations for trust enhancement.** This enhancement pattern differs from typical trust repair literature and may reflect several mechanisms specific to experimental contexts: mere exposure effects, learning and adaptation, regression artifacts, and novelty effects. These alternative processes may operate simultaneously with performance-based trust repair models. Detailed theoretical analysis of these mechanisms is provided in the Discussion.

**Additional findings.** Baseline trust in AI showed robust main effects across all outcomes (all *p*-values < 0.001, η^2^s > 0.100) but did not significantly interact with temporal dynamics, confirming its role as a stable predisposition rather than a temporal moderator. Advisory-type effects observed in Study 1 were replicated as main effects in Study 2. However, these did not significantly interact with Round (i.e., the three repeated measurement points capturing baseline, post-error, and post-repair stages), suggesting that initial algorithmic advantage persists but does not fundamentally alter error sensitivity or repair processes.

**Robustness verification.** Linear mixed-effects models with participant-level random intercepts confirmed all key findings while accounting for individual differences in baseline trust levels. The Round × Accuracy interaction remained highly significant, with Round 2 showing the most substantial accuracy effect (β = 0.947, *p* < 0.001, 95% CI [0.724, 1.171]), replicating the single-error shock pattern. The Round × Explanation interaction was robust (β = 0.678, *p* < 0.001, 95% CI [0.395, 0.962]), confirming the effectiveness of post-error repair. Financial literacy moderation was verified through significant interactions in Round × Financial Literacy (β = −0.094, *p* = 0.037, 95% CI [−0.182, −0.006]), supporting enhanced sensitivity to performance feedback. Complete LMM results are provided in Appendix A.

## 5. Discussion

### 5.1. Overview of Findings

Across two complementary experiments, this research provides the first systematic evidence for temporal trust dynamics in human–AI financial advisory interactions. Study 1 established that algorithmic advisors receive higher initial trust, satisfaction, and reliance ratings compared to human experts in financial contexts, supporting the appreciation of algorithms in consequential decision-making environments (H1). Study 2 traced the evolution of trust across repeated interactions, revealing three distinct temporal patterns that align with our theoretical framework: an initial algorithmic preference when performance uncertainty exists, acute sensitivity to single errors that violate established expectations (H2), and partial trust recovery through post-error explanations that clarify system limitations (H3). Financial literacy emerged as a critical moderator enhancing sensitivity to performance feedback across all phases (H4).

The temporal dynamics proceeded sequentially through three phases as predicted. Initial algorithm appreciation reflects users’ reliance on general competence expectations when direct performance evaluation is unavailable. A single-error shock demonstrates acute sensitivity to performance failures, with effect sizes (η^2^ > 0.100) indicating considerable practical significance. Post-error explanations facilitated meaningful but incomplete trust recovery, suggesting that transparency interventions can mitigate but not eliminate the informational value of observed failures. These patterns reveal AI trust as fundamentally cognitive-evaluative, characterized by rapid updating based on performance signals rather than gradual relationship development typical of interpersonal trust.

### 5.2. Theoretical Contributions and Integration

#### 5.2.1. Temporal Process Theory of AI Trust

This research advances trust theory by providing the first systematic temporal account of AI trust development that moves beyond static adoption models to dynamic process theories. While traditional trust frameworks ([33]; [34]) emphasize the development of stable relationships through repeated positive interactions, our findings reveal that AI trust follows fundamentally different temporal patterns, characterized by rapid formation, acute vulnerability, and targeted repair opportunities.

The three-stage framework—formation through competence cues, single-error shock, and explanation-mediated repair—establishes AI trust as a distinct cognitive-evaluative process, separate from interpersonal trust mechanisms. This distinction is theoretically significant because it suggests that AI trust operates more like performance-based confidence than relationship-based faith, making it both more volatile and more amenable to systematic intervention. The temporal specificity of these patterns provides a foundation for predictive models of human–AI interaction that can anticipate trust breakdowns and design appropriate recovery mechanisms.

#### 5.2.2. Reconciliation of Algorithm Appreciation and Aversion

Our temporal framework resolves the apparent contradiction between algorithm appreciation and algorithm aversion by demonstrating that both represent rational responses to available information at different stages of interaction. Algorithm appreciation dominates initial encounters when users lack performance feedback and must rely on general competence expectations that favor AI in analytical tasks. Algorithm aversion arises from error observations that violate established expectations, leading to a significant decline in trust despite the algorithm’s overall superior performance.

This temporal reconciliation advances behavioral decision-making theory by demonstrating that seemingly opposing biases can coexist within coherent, sequential processes, rather than representing competing psychological tendencies. The path-dependent nature of these effects—where early appreciation amplifies subsequent aversion—reveals a “trust acceleration paradox” with important implications for AI system design and user experience management.

Notably, our findings revealed unexpected trust enhancement beyond baseline levels in Round 3 across all conditions, including those experiencing errors. This enhancement effect was consistently observed across all conditions with moderate effect sizes, suggesting it reflects systematic processes rather than a statistical artifact. This pattern differs from interpersonal trust repair literature. It may reflect structured exposure effects where repeated interaction with systematic advisory processes enhances general confidence in decision support frameworks, independent of specific performance feedback. This enhancement likely reflects structured exposure to consistent advisory formats rather than mere familiarity effects, distinguishing it from the simple exposure mechanisms discussed in alternative explanations.

#### 5.2.3. Cognitive Mechanisms and Individual Differences

The financial literacy moderation effects reveal systematic heterogeneity in trust dynamics based on users’ cognitive capabilities for processing performance signals. Unlike interpersonal trust, which develops through social and emotional exchanges, AI trust calibration relies heavily on users’ analytical abilities to extract diagnostic information from system behavior and translate explanatory content into appropriate behavioral adjustments.

This cognitive foundation presents both opportunities and challenges for the deployment of inclusive AI. Users with higher domain expertise demonstrate more appropriate trust calibration through enhanced signal detection and explanation processing. However, this creates potential equity concerns where AI benefits may be systematically distributed based on user capabilities, highlighting the need for adaptive transparency systems that accommodate diverse analytical skills.

### 5.3. Alternative Explanations and Robustness Considerations

Several alternative mechanisms could potentially account for our findings. Novelty effects might explain initial AI preference, but this would predict declining algorithmic advantage across repeated interactions rather than the maintained preference we observed. Mere exposure effects could increase familiarity regardless of performance quality, but cannot account for the specific error sensitivity patterns, where single inaccuracies produced a marked decline in trust despite continued system exposure.

Regression to the mean might contribute to Round 3 recovery but would not explain systematic differences between explanation conditions or the enhancement effects across all groups. Cultural factors specific to Korean participants may influence absolute trust levels, but they would not predict the relative temporal patterns across conditions that constitute our primary contribution. Demand characteristics represent another concern; however, the complexity of interaction patterns and counterintuitive findings (such as trust enhancement in Round 3) argue against systematic response bias.

The consistency of effects across multiple measures, the specificity of hypothesized interactions, and replication through linear mixed-effects robustness checks support our theoretical interpretation over these alternatives. The LMM analyses successfully converged with participant-level random intercepts; however, more complex random slope models encountered convergence issues common in smaller samples, which limited our ability to model individual trajectory variation. Nevertheless, field experiments with actual financial stakes would provide critical external validity tests.

Additionally, manipulation checks confirmed that participants successfully distinguished between low-risk and high-risk product conditions; however, the risk level showed minimal effects across studies, consistent with exploratory treatment in prior research. This suggests that advisor type effects may override risk perceptions in early encounters; however, a systematic investigation of risk moderation represents an important future direction.

### 5.4. Cross-Domain Implications and Limitations

While our studies focused on financial advisory contexts, the temporal trust dynamics framework has broader implications for the deployment of AI across high-stakes domains. Healthcare AI systems may face similar challenges where initial algorithm appreciation establishes unrealistic expectations, making practitioners vulnerable to single-error shock when diagnostic mistakes occur. Educational AI presents parallel dynamics where initial enthusiasm might give way to rejection following observable errors. Legal AI applications in case analysis represent another domain where trust miscalibration could have serious consequences.

However, important boundary conditions may limit generalizability. Task objectivity appears crucial for algorithm appreciation; domains with subjective judgment requirements may exhibit weaker initial AI preferences ([6]). Error detectability varies across applications, with some contexts providing immediate performance feedback that could alter trust dynamics. The stakes and consequences of errors may also moderate temporal patterns, as medical errors with irreversible outcomes may fundamentally alter trust trajectories compared to recoverable financial losses.

### 5.5. Practical Implications for Design and Policy

The temporal trust dynamics framework provides concrete guidance for AI advisory systems. Initial trust formation should be supported through transparent performance disclosure that establishes realistic expectations without underselling capabilities. Error acknowledgment and explanation mechanisms represent critical requirements given acute sensitivity to single failures. Our study examined the provision of basic explanations versus their absence, and while evidence from the literature suggests that explanations combining causal attribution with boundary specification may be most effective, a systematic comparison of explanation types represents an important limitation of the present research and a priority for future investigation.

Post-error explanations should specify why the error occurred, what limitations contributed to it, and how users should interpret subsequent recommendations. Adaptive transparency strategies should take into account individual differences in processing capabilities. Users with higher domain expertise may benefit from detailed explanations, while those with lower expertise require simplified explanations that focus on behavioral implications. The single-error shock phenomenon underscores the necessity for human oversight protocols that can intervene when systems encounter failures in critical contexts.

### 5.6. Limitations and Future Research Directions

Several limitations qualify our conclusions. Both studies employed vignette-based tasks with hypothetical decisions, which may not fully capture the emotional factors present in real financial decision-making. Participants were drawn from a Korean online panel, limiting cross-cultural generalizability. Our focus on three interaction rounds over short periods cannot capture the long-term relationship development that genuine human–AI interactions involve. Additionally, reliance on self-report measures may introduce response bias or social desirability effects that could influence trust and reliance assessments.

Future research should examine error characteristics beyond single, detectable inaccuracies, including gradual degradation, ambiguous errors, and systematic biases. Explanation design requires a comprehensive investigation of different types, modalities, and timing strategies. Individual differences warrant a more comprehensive investigation, extending beyond financial literacy to encompass cognitive flexibility, technology anxiety, and cultural values. Cross-domain validation in healthcare, education, and legal contexts would clarify the scope of our theoretical framework.

Research examining actual behavioral outcomes and decision quality would provide important validation for the practical significance of trust dynamics. Understanding how trust miscalibration affects real reliance behaviors, decision accuracy, and long-term user welfare represents a critical research priority.

## 6. Conclusions

This research provides the first systematic behavioral account of how trust in AI financial advisors develops, deteriorates, and recovers through repeated interactions. Our temporal trust dynamics framework reveals that AI trust follows predictable patterns distinct from interpersonal trust: initial algorithmic advantage based on competence expectations, acute sensitivity to single errors that violate performance assumptions, and partial recovery through explanatory transparency that clarifies system limitations and provides actionable guidance. Financial literacy emerged as a critical moderator, with higher-expertise users showing enhanced sensitivity to performance feedback and more effective utilization of explanatory information across all interaction phases.

These findings have immediate implications for designing AI systems that can maintain productive human partnerships despite inevitable performance failures. Rather than pursuing perfect accuracy or generic transparency, effective AI advisory systems should focus on establishing appropriate initial expectations, providing timely explanations after errors that clarify causes and limitations, and tailoring transparency to user capabilities. For policymakers, the results underscore the need for regulatory frameworks that safeguard users lacking analytical skills, enabling them to appropriately calibrate AI trust, while ensuring that AI advisory benefits remain accessible to diverse populations.

Ultimately, this research demonstrates that successful human–AI collaboration depends not on eliminating errors, but on designing systems that can navigate the predictable dynamics of trust, error, and recovery that characterize human responses to algorithmic advice. The temporal specificity of these dynamics offers both theoretical insights into the nature of human–AI interaction and practical guidance for developing AI systems that can maintain user partnership in the face of the inevitable challenges of real-world deployment.

## Figures and Tables

**Figure 1 behavsci-15-01370-f001:**
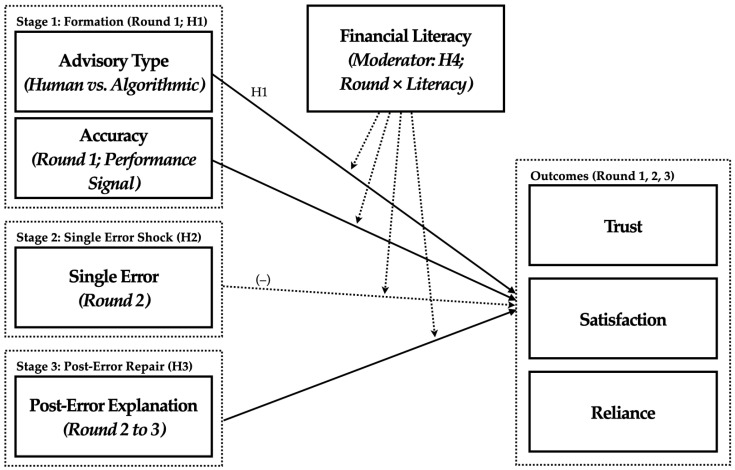
Conceptual Framework: Temporal trust dynamics with algorithmic advice (Formation → Single-Error Shock → Post-Error Repair). Solid arrows indicate hypothesized positive effects (H1: advisory type; H3: post-error explanation). Dashed arrows indicate the single-error shock and moderation (H2, H4). Baseline trust in AI is included as a covariate; product risk is examined exploratorily in Study 1. Outcomes are assessed at Rounds 1, 2, and 3 (Trust, Satisfaction, Reliance).

**Figure 2 behavsci-15-01370-f002:**
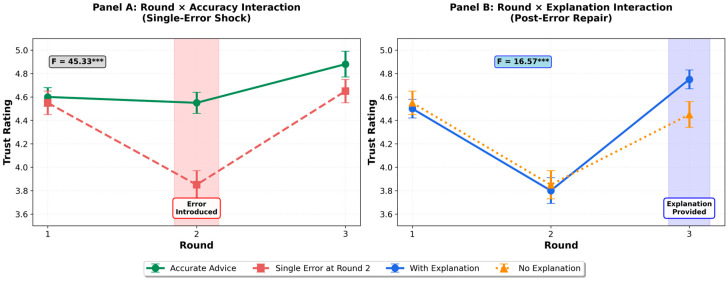
Study 2 Trust Dynamics: Single-Error Shock and Post-Error Repair. (**A**) Round × Accuracy interaction showing trust decline following single errors at Round 2 (H2). (**B**) Round × Explanation interaction showing recovery facilitated by post-error explanations (H3). Error bars represent 95% confidence intervals. *** *p* < 0.001.

**Table 1 behavsci-15-01370-t001:** Sample Characteristics. Panel A: Study 1; Panel B: Study 2.

**Panel A. Study 1**								
**Variable**	**Total Sample (N = 189)**	**Low-Risk/** **Human Expert** **(N = 43)**		**Low-Risk/** **AI Robo-Advisor** **(N = 50)**	**High-Risk/** **Human Expert** **(N = 48)**		**High-Risk/** **AI Robo-Advisor** **(N = 48)**	**X^2^/F**
Gender (%)								
Male	50.3	48.8		50.0	50.0		52.1	0.101
Female	49.7	51.2		50.0	50.0		47.9	
Age (M, SD)	39.96 (10.43)	39.19 (11.25)		40.24 (10.22)	40.08 (10.55)		40.23 (10.08)	0.102
Education (%)								
High school graduate	5.8	9.3		2.0	8.3		4.2	4.660
College graduate	82.0	79.1		90.0	77.1		81.3	
Postgraduate degree	12.2	11.6		8.0	14.6		14.6	
Household income (KRW 10 k, M, SD)	390.74 (232.72)	409.33 (267.59)		377.60 (197.33)	340.52 (202.10)		438.00 (256.59)	1.563
Risk attitude (M, SD)	5.07 (1.75)	4.60 (1.40)		5.20 (1.91)	5.38 (1.88)		5.04 (1.70)	1.610
Financial skill (M, SD)	4.69 (0.80)	4.72 (0.82)		4.66 (0.99)	4.70 (0.70)		4.69 (0.68)	0.035
Financial literacy (M, SD)	4.36 (1.47)	4.56 (1.18)		4.26 (1.61)	4.06 (1.59)		4.58 (1.38)	1.380
Self-efficacy (M, SD)	4.80 (0.88)	4.72 (1.00)		4.76 (0.96)	4.77 (0.82)		4.93 (0.75)	0.487
Investment independence (M, SD)	3.32 (0.82)	3.51 (0.74)		3.30 (0.81)	3.27 (0.74)		3.23 (0.97)	1.039
Prospectus review (%)								
Yes	37.0	30.2		38.0	41.7		37.5	1.319
No	63.0	69.8		62.0	58.3		62.5	
AI literacy (M, SD)	4.58 (0.94)	4.61 (0.72)		4.50 (1.00)	4.53 (1.02)		4.68 (0.98)	0.349
Trust in AI (M, SD)	3.97 (1.04)	3.80 (1.10)		4.08 (0.99)	3.85 (1.00)		4.12 (1.05)	1.111
**Panel B. Study 2**								
**Variable**	**Total** **Sample** **(N = 294)**	**Human/** **Accurate** **(N = 50)**	**Human/** **Inaccurate/** **+Explanation** **(N = 50)**	**Human/** **Inaccurate/** **−Explanation** **(N = 47)**	**AI advisor/** **Accurate** **(N = 50)**	**AI advisor/** **Inaccurate/** **+Explanation** **(N = 47)**	**AI advisor/** **Inaccurate/** **−Explanation** **(N = 50)**	**X^2^/F**
Gender (%)								
Male	49.7	50.0	50.0	48.9	50.0	48.9	50.0	0.029
Female	50.3	50.0	50.0	51.1	50.0	51.1	50.0	
Age (M, SD)	39.86 (10.40)	40.28 (10.14)	39.62 (10.54)	40.23 (10.26)	39.62 (10.60)	39.98 (11.56)	39.48 (9.80)	0.053
Education (%)								
High school graduate	10.9	0.0	6.0	17.0	16.0	10.6	16.0	16.059
College graduate	78.6	88.0	86.0	68.1	80.0	76.6	72.0	
Postgraduate degree	10.5	12.0	8.0	14.9	4.0	12.8	12.0	
Household income (KRW 10 k, M, SD)	409.31 (257.34)	480.00 (324.51)	438.80 (196.93)	380.64 (184.90)	385.74 (277.91)	384.47 (203.30)	383.00 (308.45)	1.285
Risk attitude (M, SD)	5.38 (2.21)	4.94 (2.32)	5.38 (1.96)	5.13 (2.21)	5.78 (2.45)	5.79 (2.07)	5.30 (2.17)	1.185
Financial skill (M, SD)	4.66 (0.89)	4.79 (0.85)	4.80 (0.90)	4.67 (0.92)	4.46 (1.01)	4.66 (0.72)	4.55 (0.89)	1.135
Financial literacy (M, SD)	4.19 (1.50)	4.28 (1.33)	4.36 (1.37)	4.21 (1.64)	4.02 (1.72)	4.06 (1.51)	4.20 (1.43)	0.360
Self-efficacy (M, SD)	4.77 (0.91)	4.96 (0.88)	4.68 (0.94)	4.89 (0.89)	4.53 (1.02)	4.77 (0.82)	4.79 (0.85)	1.406
Investment independence (M, SD)	3.40 (0.78)	3.16 (0.89)	3.44 (0.76)	3.47 (0.75)	3.44 (0.76)	3.43 (0.65)	3.50 (0.81)	1.259
Prospectus review (%)								
Yes	38.1	40.0	36.0	34.0	34.0	46.8	38.0	2.366
No	61.9	60.0	64.0	66.0	66.0	53.2	62.0	
AI literacy (M, SD)	4.60 (0.96)	4.65 (1.01)	4.70 (1.02)	4.62 (0.99)	4.30 (1.06)	4.84 (0.85)	4.49 (0.76)	1.856
Trust in AI (M, SD)	4.09 (1.02)	4.34 (0.90)	3.98 (1.08)	3.98 (1.03)	4.05 (1.03)	4.19 (1.00)	4.02 (1.09)	0.977

Notes: M = mean; SD = standard deviation; N = number of observations.

**Table 2 behavsci-15-01370-t002:** Study 1 Results: Advisory Type and Product Risk Effects.

Dependent Variable	Condition	Mean (SD)	F	*p*	η^2^
Trust	All—Human	4.44 (0.89)			
	All—AI	4.79 (0.82)			
	Low Risk—Human	4.52 (0.87)			
	Low Risk—AI	4.83 (0.94)			
	High Risk—Human	4.36 (0.92)			
	High Risk—AI	4.75 (0.69)			
	Main Effects				
	Advisory Type		7.820 **	0.006	0.041
	Risk Level		0.859	0.355	0.005
	Advisory Type × Risk		0.092	0.762	0.000
Satisfaction	All—Human	4.53 (1.15)			
	All—AI	4.89 (0.96)			
	Low Risk—Human	4.51 (1.18)			
	Low Risk—AI	4.82 (0.94)			
	High Risk—Human	4.54 (1.13)			
	High Risk—AI	4.96 (0.99)			
	Main Effects				
	Advisory Type		5.507 *	0.020	0.029
	Risk Level		0.297	0.586	0.002
	Advisory Type × Risk		0.123	0.726	0.001
Reliance	All—Human	4.12 (1.13)			
	All—AI	4.63 (0.92)			
	Low Risk—Human	3.96 (1.12)			
	Low Risk—AI	4.67 (0.90)			
	High Risk—Human	4.27 (1.13)			
	High Risk—AI	4.59 (0.96)			
	Main Effects				
	Advisory Type		11.726 ***	<0.001	0.060
	Risk Level		0.607	0.437	0.003
	Advisory Type × Risk		1.663	0.199	0.009

Notes: M = mean; SD = standard deviation; N = number of observations. * *p* < 0.05, ** *p* < 0.01, *** *p* < 0.001.

**Table 3 behavsci-15-01370-t003:** Study 2 Repeated-Measures ANOVA Results Panel A: Round × Accuracy Interactions (H2: Single-Error Shock) Panel B: Round × Explanation Interactions (H3: Post-Error Repair).

**Panel A. Round × Accuracy**									
	**Trust**			**Satisfaction**			**Reliance**		
	**df**	**F**	**Partial η^2^**	**df**	**F**	**Partial η^2^**	**df**	**F**	**Partial η^2^**
**Between-Subject Effects**									
Gender (male = 0)	1	0.001	0.000	1	0.793	0.003	1	0.515	0.002
Age	1	1.557	0.006	1	0.111	0.000	1	0.299	0.001
Monthly income (KRW 10 k)	1	0.288	0.001	1	0.238	0.001	1	1.121	0.004
High school (university = 0)	1	0.338	0.001	1	0.375	0.001	1	0.268	0.001
Graduate (university = 0)	1	1.510	0.005	1	1.682	0.006	1	1.039	0.004
Risk Attitude	1	0.051	0.000	1	0.054	0.000	1	0.018	0.000
Financial Skill	1	0.009	0.000	1	0.179	0.001	1	0.126	0.000
Financial Literacy	1	0.869	0.003	1	0.575	0.002	1	2.421	0.009
Self-efficacy	1	1.518	0.005	1	0.260	0.001	1	0.000	0.000
Investment independence	1	1.237	0.004	1	0.249	0.001	1	0.391	0.001
Prospectus review (don’t = 0)	1	0.020	0.000	1	0.550	0.002	1	1.016	0.004
AI Literacy	1	0.030	0.000	1	0.094	0.000	1	0.000	0.000
Trust in AI	1	45.558 ***	0.141	1	32.136 ***	0.104	1	47.465 ***	0.146
Advisory Type	1	5.650 *	0.020	1	4.236 *	0.015	1	3.507	0.013
Accuracy	1	14.035 ***	0.048	1	10.081 **	0.035	1	6.568 *	0.023
Advisory Type × Accuracy	1	0.368	0.001	1	0.018	0.000	1	0.027	0.000
**Within-Subjects Effects**									
Round (Time Effect)	1.916	0.181	0.001	1.928	0.701	0.003	1.910	0.132	0.000
Round × Gender (male = 0)	1.916	0.532	0.002	1.928	0.002	0.000	1.910	0.096	0.000
Round × Age	1.916	1.329	0.005	1.928	1.160	0.004	1.910	1.757	0.006
Round × Monthly income (KRW 10 k)	1.916	1.710	0.006	1.928	0.693	0.002	1.910	1.586	0.006
Round × High school (university = 0)	1.916	0.614	0.002	1.928	1.969	0.007	1.910	1.325	0.005
Round × Graduate (university = 0)	1.916	1.167	0.004	1.928	0.316	0.001	1.910	0.575	0.002
Round × Risk attitude	1.916	1.921	0.007	1.928	3.552 *	0.013	1.910	0.887	0.003
Round × Financial skill	1.916	1.122	0.004	1.928	1.598	0.006	1.910	1.181	0.004
Round × Financial literacy	1.916	4.126 *	0.015	1.928	2.100	0.008	1.910	3.527 *	0.013
Round × Self-efficacy	1.916	1.005	0.004	1.928	0.978	0.004	1.910	0.877	0.003
Round × Investment independence	1.916	4.777 *	0.017	1.928	6.376 **	0.022	1.910	4.293 *	0.015
Round × Prospectus review (don’t = 0)	1.916	2.742	0.010	1.928	2.141	0.008	1.910	0.648	0.002
Round × AI literacy	1.916	0.498	0.002	1.928	0.046	0.000	1.910	0.847	0.003
Round × Trust in AI	1.916	1.612	0.006	1.928	0.675	0.002	1.910	0.541	0.002
Round × Advisory Type	1.916	0.711	0.003	1.928	1.210	0.004	1.910	1.096	0.004
Round × Accuracy	1.916	45.327 ***	0.141	1.928	33.015 ***	0.106	1.910	36.600 ***	0.117
Round × Advisory Type × Accuracy	1.916	1.661	0.006	1.928	1.189	0.004	1.910	0.731	0.003
**Panel B. Round × Explanation**									
	**Trust**			**Satisfaction**			**Reliance**		
	**df**	**F**	**Partial η^2^**	**df**	**F**	**Partial η^2^**	**df**	**F**	**Partial η^2^**
**Between-Subject Effects**									
Gender (male = 0)	1	0.007	0.000	1	1.690	0.009	1	1.006	0.006
Age	1	1.853	0.010	1	0.355	0.002	1	0.228	0.001
Monthly income (KRW 10 k)	1	0.137	0.001	1	0.007	0.000	1	0.397	0.002
High school (university = 0)	1	0.086	0.000	1	0.028	0.000	1	0.042	0.000
Graduate (university = 0)	1	5.317 *	0.029	1	3.275	0.018	1	4.858 *	0.027
Risk Attitude	1	0.044	0.000	1	0.000	0.000	1	0.030	0.000
Financial Skill	1	0.258	0.001	1	0.001	0.000	1	0.063	0.000
Financial Literacy	1	1.221	0.007	1	0.511	0.003	1	1.711	0.010
Self-efficacy	1	1.247	0.007	1	0.000	0.000	1	0.191	0.001
Investment independence	1	1.881	0.011	1	0.136	0.001	1	0.018	0.000
Prospectus review (don’t = 0)	1	0.076	0.000	1	0.019	0.000	1	0.313	0.002
AI Literacy	1	0.151	0.001	1	0.010	0.000	1	0.039	0.000
Trust in AI	1	22.085 ***	0.111	1	15.230 ***	0.079	1	31.144 ***	0.150
Advisory Type	1	3.030	0.017	1	4.471 *	0.025	1	2.913	0.016
Explanation	1	2.505	0.014	1	1.569	0.009	1	1.580	0.009
Advisory Type × Explanation	1	0.222	0.001	1	0.019	0.000	1	0.194	0.001
**Within-Subjects Effects**									
Round (Time Effect)	1.869	0.721	0.004	1.839	1.555	0.009	1.805	0.398	0.002
Round × Gender (male = 0)	1.869	1.264	0.007	1.839	0.214	0.001	1.805	0.154	0.001
Round × Age	1.869	0.419	0.002	1.839	0.007	0.000	1.805	0.595	0.003
Round × Monthly income (KRW 10 k)	1.869	0.747	0.004	1.839	0.454	0.003	1.805	0.682	0.004
Round × High school (university = 0)	1.869	0.081	0.000	1.839	1.950	0.011	1.805	1.541	0.009
Round × Graduate (university = 0)	1.869	0.864	0.005	1.839	0.348	0.002	1.805	0.365	0.002
Round × Risk attitude	1.869	2.931	0.016	1.839	2.152	0.012	1.805	1.194	0.007
Round × Financial skill	1.869	2.423	0.014	1.839	1.879	0.011	1.805	2.196	0.012
Round × Financial literacy	1.869	4.048 *	0.022	1.839	2.674	0.015	1.805	3.638 *	0.020
Round × Self-efficacy	1.869	1.092	0.006	1.839	0.076	0.000	1.805	0.141	0.001
Round × Investment independence	1.869	6.085 **	0.033	1.839	6.329 **	0.035	1.805	3.566 *	0.020
Round × Prospectus review (don’t = 0)	1.869	3.526 *	0.020	1.839	3.073	0.017	1.805	1.054	0.006
Round × AI literacy	1.869	1.277	0.007	1.839	1.933	0.011	1.805	1.119	0.006
Round × Trust in AI	1.869	2.877	0.016	1.839	0.873	0.005	1.805	0.162	0.001
Round × Advisory Type	1.869	1.378	0.008	1.839	1.952	0.011	1.805	0.976	0.005
Round × Explanation	1.869	16.573 ***	0.086	1.839	6.709 **	0.037	1.805	4.079 *	0.023
Round × Advisory Type × Explanation	1.869	0.836	0.005	1.839	2.221	0.012	1.805	3.533 *	0.020

Notes: Significance level: * *p* < 0.05, ** *p* < 0.01, *** *p* < 0.001.

## Data Availability

The data presented in this study are available on request from the corresponding author.

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
