# Peer review of "Trust Formation, Error Impact, and Repair in Human–AI Financial Advisory: A Dynamic Behavioral Analysis"

_behavsci, 2025, doi:10.3390/bs15101370_

Round 1
Reviewer 1 Report
Comments and Suggestions for Authors
The research question is distinctive and important, but the manuscript requires stronger theoretical grounding, clearer empirical logic, and more rigorous data analysis to substantiate its conclusions.
Major points for revision:
1. Theory and competing predictions. The authors acknowledge both algorithm aversion and algorithm appreciation in the literature, yet the paper ends up presenting evidence only for algorithm appreciation. The manuscript should (a) explain more clearly why algorithm appreciation is the expected outcome in this context, and (b) reconcile this choice with prior findings showing aversion. Furthermore, the authors mention potential moderators (e.g., cognitive load) as possible boundary conditions, but the present studies do not test or falsify these moderators. Either incorporate tests of these moderators or explicitly state that doing so is left for future work and clarify how this omission limits interpretation.
2. Manipulation/check of risk level and discrepant findings. Across studies the only consistent main effect is advisory type; no effects of risk level are reported. This pattern contradicts many prior findings in the domain. The authors should (a) justify the absence of risk effects theoretically, or (b) examine whether the risk manipulation failed (manipulation check statistics, effect sizes, descriptive differences). If the manipulation was weak, revise materials or reanalyze using a manipulation-strength index.
3. Statistical reporting, analysis choices, and covariates (Study 2). Statistical reporting is unclear in several places. For example, multiple-comparison procedures are invoked but F tests are reported without degrees of freedom or adequate detail, making it impossible for readers to evaluate the tests. Please report full test statistics (test type, test statistic, df, p-value, effect size, CIs). Moreover, Financial literacy appears conceptually continuous. The authors must justify why they analyzed it via ANOVA/F-tests (categorization) rather than treating it as a continuous moderator in regression or mixed-effects models. Further, the analytic strategy should be strengthened: linear mixed-effects models (subject-level random intercepts and slopes) are more appropriate for repeated measures / round-based data and will better model within-person dependency and time trends. Please re-run key tests with LMMs and report the results alongside the original analyses. Additionally, important covariates (demographics, prior experience such as driving or domain experience) were not controlled; adding these controls could change the results. Report analyses with these covariates included, or justify their exclusion.
4. Figures for Study 2 are poorly formatted and difficult to interpret (unclear legends, line styles, and annotations). Replot with clear legends, error bars (e.g., ±SE or 95% CI), and labeled axes.
5. Interpretation of repair effects, and surprising Round-3 pattern. The pattern in Round 3 is unexpected: results appear to recover fully to the accurate baseline across conditions. This is surprising because prior literature typically finds incomplete trust repair. The manuscript currently implies that transparency or explanation drives repair, but the presented pattern suggests explanation may not matter in Round 3. The authors should (a) operationalize “trust repair” clearly (e.g., change from Round1 → Round3 vs. Round1 → Round2), (b) test the repair contrast directly (Round3 vs. Round1) with proper statistics and effect sizes, and (c) offer a plausible reason for full recovery (if robust).
6. The manuscript’s logic is unclear: Study 1 reads like a pilot/pretest, and most research questions seem addressable within Study 2 alone. Clarify the purpose of Study 1, and explain explicitly how Study 1 uniquely contributes beyond Study 2. If Study 1 is merely a pilot, label it as such and streamline presentation accordingly.
Reviewer 2 Report
Comments and Suggestions for Authors
To authors
This manuscript examines the development of trust in human–AI interaction, formation, error, and repair, within the context of financial advice. The focus on temporal dynamics is timely. However, to reach its full potential, please consider addressing the following comments:
Introduction
The introduction sets up the problem adequately, but the theoretical framework feels underdeveloped. The paper lists several relevant theories (e.g., trait activation theory, perfect automation schema), but does not sufficiently justify the model selected.
The reason for focusing on finance is clear, but the move to “implications beyond finance” is abrupt. Adding a short paragraph comparing finance to other high-stakes areas (such as education) would strengthen the argument.
Literature Review
The link between the reviewed literature and the specific hypotheses needs to be strengthened. At times, the hypotheses appear to be plausible statements rather than propositions derived directly and rigorously from the theoretical discussion.
H2 (Single-Error Shock): The concept of a "perfect automation schema" is central to your argument, yet it is not adequately justified. Is this a universal schema? Does it vary by culture, age, or prior technology experience? The manuscript would be substantially strengthened by providing more evidence for the prevalence of this schema and discussing potential boundary conditions. As it stands, it is a strong assumption that requires more support.
H3 (Repair via Explanation): The hypothesis posits that explanations can repair trust. However, the literature on Explainable AI (XAI) is complex; not all explanations are adequate. The hypothesis needs to be more specific. What type of explanation (e.g., mechanistic, why-not, counterfactual) is expected to be effective and why? Consider providing a stronger theoretical rationale for why the chosen form of explanation in your study should work.
Methods
This section requires clarification and justification. An experimental design's validity depends on the details of its implementation, and several key details are currently insufficiently justified.
Ecological Validity: The simulation of a "financial advisory" context is a critical point of concern. How realistic was this simulation? Participants made hypothetical choices with no real-world consequences. This limitation is expected to be more prominently acknowledged and discussed. Please provide a more detailed description of the experimental task to allow readers to judge its realism. Why was this specific task chosen over others?
Stimuli Design: Please provide the exact wording for the experimental manipulations, especially the error messages and the post-hoc explanations. These are the core of your study. Without them, it is very difficult for reviewers or readers to assess their validity.
Results
The Results section presents the statistical outputs of your experiments. However, the presentation currently lacks clarity and narrative coherence, making it somewhat difficult for the reader to follow the logical thread of your findings.
Although the tables provide full statistical information (e.g., F, df, η²), the narrative presentation remains somewhat fragmented. I recommend more clearly integrating these statistics into the text so that the results are directly linked to each hypothesis and easier for readers to follow.
Structure and Narrative Flow: The current structure reads like a direct output from statistical software. The section needs to be reorganised around your hypotheses. For each hypothesis, you should:
- Clearly state which hypothesis is being tested.
- Direct the reader to the specific analysis used to test it.
- Present the statistical results of that analysis in a clear, concise sentence.
- State explicitly whether the results support or do not support the hypothesis.
For example, instead of writing "An ANOVA was conducted on trust scores," consider rewriting as, "To test H1, which predicted that participants would show higher initial trust in the AI advisor, we conducted a one-way ANOVA. The results supported this hypothesis, showing a significant main effect for advisor type, F(df, df) = X.XX, p < .001, η² = .XX, with the AI advisor (M = X.X, SD = X.X) rated significantly higher than the human advisor (M = X.X, SD = X.X)."
Discussion (pp. 16–18)
The discussion section should move beyond restating the results. A deeper interpretation is needed, including a more critical self-reflection on the study's limitations and the plausibility of alternative explanations for your findings. For example, could the results be explained by factors other than your proposed framework? Could the initial preference for the AI advisor be due to novelty effects rather than "algorithm appreciation"? Could the trust decline be a simple regression to the mean?
In lines 912–915, the manuscript points out that future research should investigate whether the temporal trust patterns can be found in other domains with different levels of risk and decision-making stakes. I agree this is an important direction. While healthcare and law are often discussed as high-stakes areas, education may also provide useful illustrations.
Conclusion
This section is good.
Round 2
Reviewer 2 Report
Comments and Suggestions for Authors
Thank you for your careful and comprehensive revision. I am very happy to see how thoroughly you engaged with the feedback from the initial review. Overall, the manuscript has been considerably strengthened and now presents a sound contribution to understanding temporal trust dynamics in human–AI financial advisory contexts. I have no further concerns and recommend the paper for publication.